# Influences of Particle Size and Addition Level on the Rheological Properties and Water Mobility of Purple Sweet Potato Dough

**DOI:** 10.3390/foods12020398

**Published:** 2023-01-13

**Authors:** Han Hu, Xiangyu Zhou, Yuxin Zhang, Wenhua Zhou, Lin Zhang

**Affiliations:** 1Hunan Key Laboratory of Processed Food for Special Medical Purpose, National Engineering Research Center of Rice and Byproduct Deep Processing, School of Food Science and Technology, Central South University of Forestry & Technology, Changsha 410004, China; 2Division of Medicine, Faculty of Medical Science, University College London, London WC1E 6BT, UK

**Keywords:** purple sweet potato, dough, particle size, rheological property, water mobility

## Abstract

This paper investigated the effects of different particle sizes and addition levels of purple sweet potato flour (PSPF) on the rheological properties and moisture states of wheat dough. There was deterioration in the pasting and mixing properties of the dough, due to the addition of PSPF (0~20% substitution), which was reduced by decreasing the particle size of the PSPF (260~59 μm). Dynamic rheology results showed that PSPF enhanced the elasticity of the dough, providing it solid-like processability. PSPF promoted the binding of gluten proteins and starch in the dough, resulting in a denser microstructure. Differential scanning calorimetry and low-field nuclear magnetic resonance showed that PSPF converted immobilized water and freezable water to bound water and non-freezable water in the dough, making the dough more stable, and that the reduction in PSPF particle size facilitated these processes. Our results provide evidence for the great application potential of purple sweet potatoes for use in flour-based products.

## 1. Introduction

Purple sweet potato (*Ipomoea batatas*) (PSP) is a special variety of sweet potatoe, rich in anthocyanins, a group of flavonoids with a benzopyrene structure, which confers a bright purple colour and health benefits [1]. Due to its high anthocyanin content, PSP is typically prepared as purple sweet potato flour (PSPF) and added to wheat or rice flours to provide products with a unique colour, flavour, and nutrition. Cui and Zhu found that Chinese steamed bread prepared with PSPF has enhanced colour, polyphenol content, and antioxidant activity [2]. Wang et al. reported that PSP improved the structural properties of rice products [3].

Conversely, a lack of gluten- and alcohol-soluble protein in most exogenous flours, including PSPF, is often detrimental to the rheological properties [4]. Several studies have reported various common pre-treatments of raw materials, including pre-gelatinization, degreasing, inactivation, and fermentation [5,6,7,8], as well as the addition of food additives such as proteases, emulsifiers, and edible gums [9,10]. However, these approaches were limited by the reduced nutritional values of raw materials and residual additives. Thus, it may be more innovative, environmentally friendly, and achievable to change the particle size and addition level of the non-gluten flours. Changes in the particle size of the flour affect its specific surface area and porosity, which in turn affects the packing density, the number of binding sites, and the exposure of active ingredients of the flour, leading to an impact on both the formation and nutritional value of the dough [11]. Doughs containing superfine bran have higher water absorption and gelatinization temperatures, peak viscosity, final viscosity, and setback value than dough containing coarse bran [12]. Meanwhile, Azeem, Mu, and Zhang reported that doughs prepared from the smaller particle size of orange-fleshed sweet potato flour had a higher maximum height during fermentation and exhibit optimal elastic and viscous modulus values [13]. However, few studies have investigated the influences of the particle size and addition level of PSPF on the doughs.

Furthermore, the moisture status in the food system also has an important influence on the processing and storage characteristics of the food [14]. The process of dough formation is mainly dependent on the combinations of water molecules and macromolecules, such as proteins and starch, in the dough. The water used to hydrate the flour components and form the gluten network structure is called “bound” water. The “bound” and “free” water play an important role in the stability of the dough [15]. Due to many hydrophilic groups in the PSPF fibres, it is expected that doughs prepared by the partial substitution of PSPF will have a different moisture distribution to wheat dough. However, there are no in-depth studies on this topic reported so far.

Therefore, this study aimed to incorporate the different particle sizes and addition levels of PSPFs into the doughs and further investigate the pasting properties, mixing properties, dynamic rheological properties, characterizations of the moisture state, and water mobility of this dough. This information will help in the development of high-quality flour-based products enriched with PSP in the future.

## 2. Materials and Methods

### 2.1. Materials

The PSP variety Mianzishu 9 was supplied by the Hunan Academy of Agricultural Sciences (Changsha, China) and planted in the experimental field of the Hunan Crop Research Institute (Changsha, China). After harvesting in October 2022, fresh PSPs, free of rot, sprouting, diseases, and pests were selected and immediately sent to our laboratory. The PSPs were cleaned and cut into small cubes. The tubers were then cooked, peeled, and dehydrated in a cabinet dryer (101-3EBS, Yongguang Medical Instruments Co., Ltd., Beijing, China) at 60 °C. A tuber was removed, and its moisture content was determined every half hour using a halogen rapid moisture analyser (JH-HS, Yixind Instrument Co., Ltd., Taizhou, China), until the moisture was maintained at 6–10%. The dried PSP tubers were crushed into powder in a pulveriser (FW-400A, Zhongxing Weiye Instrument Co., Ltd., Beijing, China) at a speed of 26,000 r/min. The resulting flour was successively passed through 180 mesh, 140 mesh, and 100 mesh sieves to obtain three kinds of PSPFs with different particle sizes and set aside. Wheat flours were purchased from Kaixue Grain & Oil Food Co., Ltd. (Changsha, China). The moisture and protein content of PSPFs and wheat flours were measured with a halogen rapid moisture analyser (JH-HS, Yixind Instrument Co., Ltd., Taizhou, China) and a Kjeltec K9840 analyser (Haineng Instruments Co., Ltd., Jinan, China). The starch and cellulose contents were measured with a starch assay kit and a cellulose assay kit, which were purchased from Solarbio Technology Co., Ltd., Beijing, China.

### 2.2. Particle Size Distributions

A particle size analyser (LS 13320, Beckman Instruments, Inc., Fullerton, CA, USA) was used to determine the particle size of wheat flour and PSPF, following the method of Wang et al. [16], and the sizes are presented as Sauter mean diameter (D_4.3_, µm).

### 2.3. Preparation of the Formulated Flour

Wheat flour (WF) was formulated by substituting with 0%, 10%, 15%, and 20% PSPF. After thorough mixing, 10 kinds of formulated flours (FF) were obtained, which were WF, FF-10l (10% PSPF sieved through 100 mesh sieves, 90% WF), FF-10m (10% PSPF sieved through 140 mesh sieves, 90% WF), FF-10s (10% PSPF sieved through 180 mesh sieves, 90% WF), FF-15l (15% PSPF sieved through 100 mesh sieves, 85% WF), FF-15m (15% PSPF sieved through 140 mesh sieves, 85% WF), FF-15s (15% PSPF sieved through 180 mesh sieves, 85% WF), FF-20l (20% PSPF sieved through 100 mesh sieves, 80% WF), FF-20m (20% PSPF sieved through 140 mesh sieves, 80% WF), and FF-20s (20% PSPF sieved through 180 mesh sieves, 80% WF). These formulated flours were collected in polyethylene bags, vacuum sealed by a heat sealer, and stored at 4 °C until use.

### 2.4. Pasting Properties

The pasting properties of the formulated flours were investigated using a Rapid Visco-Analyzer (RVA-Super 4, Newport Scientific, Jessup, MD, USA), according to a method by Yang et al. [17], with slight modifications. Briefly, 3.5 g of the sample (14% wet basis moisture correction) was weighed into a measuring cylinder, and 25 mL of distilled water was added. The mixture was stirred until no obvious powder clumps were visible, before being placed in the RVA rotating hydrazine. The initial temperature of the RVA was set to 50 °C for 1 min. The temperature was increased to 95 °C at a rate of 12 °C per min for 2.5 min, and then cooled down to 50 °C at the same rate for 2 min, for a total duration of 13 min. The RVA pasting curve provides parameters such as peak viscosity, holding viscosity, final viscosity, breakdown, and setback of the recipe flours. The experiment was conducted with three replicates of each sample.

### 2.5. Mixing Properties

The mixing properties of the formulated flours were investigated using a Micro-dough LAB (Micro-dough LAB 2800, Dachol Technologies, Stockholm, Sweden), based on a method by Li et al. [18], with slight modifications. The water absorption was determined to reach a peak of 500 BU using a 3.50 ± 0.01 g sample at a 14% moisture basis after moisture correction. For 15 min, samples were mixed at a speed of 63 rpm and at a temperature of 30 °C. Dough development time, dough stabilisation time, and bandwidth were calculated using the mixing curve. The experiment was conducted with three replicates of each sample.

### 2.6. Dynamic Rheological Properties

The dough dynamic rheological properties were measured using a rheometer (DHR-2, TA Instruments, New Castle, DE, USA), following Mironeasa et al.’s [19] method, with slight modifications. Briefly, 25 g of the sample was weighed and placed in the Micro-dough LAB. The wheat flour dough (WFD) and purple sweet potato dough (PSPD) was removed and wrapped in plastic wrap when maximum consistency was reached. The doughs were marked as PSPD-10l–20s, depending on the type of FF used. The doughs were then placed on the rheometer test bench; excess dough was scraped off, and the dough was left to stand at 25 °C for 5 min. Test conditions were as follows: plate diameter 40 mm, strain 2.0%, scanning frequency 0.1–20.0 Hz, and plate gap 1000 μm. The experimental data acquired are described by the power law model [19,20]:(1)$$G^{\prime }=K^{\prime }\times \omega^{n\prime }\;$$
(2)$$G^{\prime \prime }=K^{\prime \prime }\times \omega^{n\prime \prime }$$
where G′ is the storage modulus (Pa); G″ is the loss modulus (Pa); ω is the angular frequency (rad/s); and K′, K″ (Pa·sn’), n′, and n″ are the experimental constants.

### 2.7. Scanning Electron Microscopy (SEM)

The microstructure of the dough was observed using an SEM with reference to Zheng’s method [20]. The doughs were lyophilised and fractured by folding. The doughs were then fixed on the sample stage using conductive double-sided tape, and a thin layer of gold was sputtered on the cross-section of the sample. The 500× cross-sectional images were observed with SEM (TESCAN MIRA LMS, Tescan Co., Ltd., Brno, Czech Republic) at an accelerating voltage of 10 kV.

### 2.8. Differential Scanning Calorimetry (DSC)

Referring to He et al.’s [21] approach, a differential scanning calorimeter (TA Instruments, Waters Technology, New Castle, DE, USA) was used to measure the enthalpy (*ΔH*) of the melting peak in the treated samples. A slice subsample of 15–30 mg from each sample was placed into the DSC pan. The pan was hermetically sealed and cooled from 25 °C to −40 °C at 5 ℃/min using liquid nitrogen and then heated to 40 °C at 5 ℃/min. ΔH of the melting peak was determined with Universal Analysis software (TA Instruments, Waters Technology, New Castle, DE, USA). The freezable water (FzW) content was calculated using the following formula:(3)$$\mathrm{FzW}\;\left(\%\right)=\Delta H/\left({\Delta H}_{0}\times {\;W}_{t}\right)\times 100$$
where FzW is the freezable water content, ΔH is the enthalpy of the melting peak of the endothermic curve, J/g, ΔH_0_ is the enthalpy of melting peak of pure water, 334 J/g, and W_t_ is the total water content of the sample. Each measurement was performed in triplicate.

### 2.9. Low-Field ^1^H Nuclear Magnetic Resonance (LF-NMR)

To evaluate the effect of PSPF on water mobility and distribution in the dough, transverse relaxation time (T_2_) was recorded using a LF-NMR (NMI120, Niumag Electronic Technology Co., Ltd., Shanghai, China). The preparation of the dough was the same as in Section 2.6. The instrument was equipped with a 60 mm probe. The relaxation curves were acquired using a Carr–Purcell–Meiboom–Gill (CPMG) pulse sequence. Data from 2000 echoes were obtained as eight-scanned repetitions. The repetition time between the two successive scans was set at 1 s [22]. All tests were performed in triplicate.

### 2.10. Statistical Analysis

The results are expressed as mean ± standard deviations. Statistical differences of all the samples were performed by two-way analysis of variance (ANOVA) and Duncan’s multiple range test (*p* < 0.05) using SPSS 16.0 (SPSS Inc., Chicago, IL, USA). The graphs were plotted using Origin 9.0 (Origin Lab Corporation, Northampton, MA, USA).

## 3. Results and Discussion

### 3.1. Particle Size Distribution and Nutrient Distribution

The particle size distribution is important for the physicochemical and functional properties of the flour [23]. As shown in Table 1, unscreened WF had a particle size of 80 μm. PSPF was sieved through 180, 140, and 100 mesh sieves, leading to D_4,3_ values of 59, 117, and 260 μm, respectively. The moisture content of PSPF significantly (*p* < 0.05) increased with decreasing particle size. This is because PSPF with a lower particle size has a larger specific surface area and is more capable of adsorbing water [24]. The contents of protein, cellulose, and starch in PSPF significantly (*p* < 0.05) decreased with decreasing PSPF particle size, and macronutrients were destroyed during mechanical crushing. The lower the particle size, the higher the degree of destruction. These findings are consistent with those of Azeem, Mu, and Zhang [13], who reported a similar relationship between particle size and the nutrient content of orange-fleshed sweet potato flour.

### 3.2. Pasting Properties

Table 2 summarized the influence of the addition level and particle size of PSPF on the pasting properties of dough. The results showed that the peak viscosity (PV), holding viscosity (HV), and final viscosity (FV) of the doughs decreased significantly (*p* < 0.05) with the increase in PSPF addition. Substitution of wheat flour with PSPF resulted in a reduction in the cross-linking of the starch structure in the blend paste, resulting in a weakening of the paste viscosity [25]. However, to some extent, the decrease in viscosity due to the addition of PSPF could be offset by the decrease in PSPF particle size. The PV, HV, and FV of the doughs increased significantly (*p* < 0.05) with the decrease in PSPF particle size. As the particle size decreased, more of the starch in PSPF was converted into more water-absorbent damaged starch through high intensity crushing, thus enhancing gelatinization [26] and increasing viscosity. Breakdown (BD) is related to the stiffness of the expanded starch granules and captures the stability of the dough during pasting. The lower the BD value, the higher the thermal stability of the dough.

As shown in Table 2, the BD of the doughs decreased significantly (*p* < 0.05) as the amount of PSPF substitution increased from 0 to 20%. This may be due to the hydroxyl groups in the starch structure of PSP interacting with large amounts of water. Thus, wheat starch was more stable during water swelling, and the thermal stability of the dough was enhanced [27]. The BD values of the doughs increased significantly (*p* < 0.05) with the decrease in PSPF particle size. This may be due to the worsening thermal stability of the blend paste with smaller PSPF particles. Setback (SB) is an index of the retrogradation tendency of amylose. Table 2 reported that the SB of the doughs decreased significantly (*p* < 0.05) as the concentration of PSPF increased from 0 to 20%, which may be explained by the high content of protein and cellulose in PSPF, which inhibits the rearrangement of starch hydrogen bonds and the retrogradation of starch [28]. With the decrease in PSPF particle size, the SB of the doughs increased significantly (*p* < 0.05).

### 3.3. Mixing Properties

The addition level and particle size of PSPF significantly affected the mixing properties of the dough, and these effects were detected on the Micro-dough LAB. Water absorption (WA) is a key indicator of the formation of dough from flour [29]. Table 3 shows that the WA of the doughs increased significantly (*p* < 0.05) as the addition level increased and the particle size of PSPF decreased, because the cellulose of PSPF contains more hydrophilic groups. Meanwhile, the smaller the PSPF particles, the more hydrophilic groups are exposed [24].

The dough development time (DT) and stabilisation time (ST) reflect the speed of the gluten network formation and the mechanical resistance of the dough, respectively. With the substitution of PSPF increasing from 0% to 20%, DT increased, and ST decreased slightly. This is probably because PSPF substitution might dilute the gluten proteins in the dough, hinder the development of the gluten network [30], and increase the kneading resistance of the dough [27]. Furthermore, doughs substituted with medium-sized PSPF particles (FF-m) had higher DT values than those substituted with small (FF-s) or large (FF-l) PSPF particles. This might be attributed to the different contents of PSPF protein and starch with different particle sizes, and the different gel systems formed alternately with the gluten network, thus resulting in different formation speeds of the gluten network [26]. However, the bandwidth (Bw), representing the elastic properties of the dough, followed a different pattern. Regarding addition amount, FF-20 showed the narrowest Bw. In terms of particle size, the Bw of FF-l was higher than that of the other samples. Variations in Bw data in different doughs might be due to the differential swelling of starch granules [31].

### 3.4. Dynamic Rheological Properties

Dynamic rheological properties are used to measure the viscoelasticity of doughs and to determine their textures and processing characteristics. Figure 1 shows the mechanical spectra of the WFD and PSPD. As the addition level of PSPF in the wheat flour increased from 0% to 20%, both G′ and G″ moduli values increased. The increase in the dynamic moduli may be due to the high water absorption of PSPF fibres, allowing PSPF to bind more tightly to gluten proteins and starch, facilitating gluten aggregation and producing more elastic behaviour [32]. Similar variations were found in dough substituted with tomato seed flour [33]. The reduction in PSPF particle size also resulted in an increase in the moduli values of the doughs, which may be explained by the smaller PSPF particles having larger specific surface areas and more hydrophilic groups exposed, resulting in a greater ability for water adsorption. Moreover, G′ values were higher than G″ in all the dough samples, indicating that the elasticity of PSPD was dominant and that PSPD could be considered for food processing [33].

The parameters of the power law equations were used to describe the dependence of the moduli on the oscillation frequency, which are presented in Table 4. The dependency of G′ and G″ dynamic moduli on the oscillation frequency was well modelled by Equations (1) and (2) over the test frequency range of 1–20 Hz. The correlation coefficients were higher than 0.996 for G′ and 0.965 for G″, showing that the power law model was adequate for modelling the viscoelastic properties of PSPD. K′ and K″ values increased with the increase in PSPF addition levels, indicating that the addition of PSPF resulted in stiffer and more elastic dough. PSPD-10 had a relatively soft texture and might be suitable for use in edible products such as bread, pizza, and pies [34,35]. PSPD-20 was more elastic and might be suitable for chewy products, such as noodles [34,36]. The higher K′ and K′ of the smaller particle size PSPD may be attributed to the finer PSPF filling the gluten network better, ensuring the integrity of the gluten network to a greater extent and giving the dough a soft and stable state. The results obtained with the G′ slope expressed in n′ presented lower values of n′ than those of the G″ slope. This is because PSPF can bind to gluten proteins through ionic bonds, thereby enhancing the viscoelasticity of the dough. The n′ values increased by increasing the addition of PSPF and decreasing the PSPF particle size.

### 3.5. Microstructure

The microstructure of doughs greatly affects their rheological behaviour. In this study, SEM was used to investigate the effect of PSPF particle size and addition level on the cross-sectional microstructure of doughs. As shown in Figure 2, a typical starch–gluten protein microstructure was observed in WFD doughs prepared from pure wheat flour. The starch and gluten proteins were relatively well ordered together, and the structure was relatively stable. With the substitution of PSPF, holes appeared in the dough structure, probably because the PSPF prevented the formation of the original structure. PSPD with the addition of smaller particle size PSPF had a more compact structure because the smaller particle size PSPF was able to fill the gaps between the gluten proteins and starch effectively.

When the PSPF addition level rose to 15% and 20%, the dough structure became full and dense, making it difficult to observe the original structure. This might be related to the high water absorption of the dough at high PSPF addition levels. The dough absorbed more water during development, and the hydrophilic groups in the system promoted the inter-binding of gluten proteins, starch, and PSPF, making them tightly connected. This result supports the dynamic rheology findings that higher additions and lower particle sizes of PSPF resulted in stiffer and more elastic dough textures. However, further investigations were required regarding how PSPF influences the water state in dough.

### 3.6. DSC

Table 5 shows the effects of different particle sizes and the addition levels of PSPF on the ΔH, freezable water (FzW), and non-freezable water (NFW) contents of the dough. The different addition levels of PSPFs significantly (*p* < 0.05) reduced the ΔH and FzW content in the dough. This indicates that the distribution of water in the dough was altered by the addition of PSPF, resulting in the migration and redistribution of water molecules throughout the system. This migration behaviour became more pronounced with increasing amounts of PSPF added. This was primarily caused by the large number of hydroxyl groups in PSPF fibres, which are exposed when dissolved in aqueous solutions and can interact with water molecules through hydrogen bonding, leading to a decrease in the mobility of water molecules and the FzW content of the dough [37]. In contrast, the addition of PSPF increased the viscosity of the aqueous solution, thus changing the rheological properties of the dough and limiting the movement of FzW molecules in the dough, thus reducing the FzW content. At a fixed level of substitution, as the particle size of PSPF decreased, the specific surface area increased, and more hydrophilic groups were exposed. The mobility of water molecules in the dough and the FzW content were further decreased.

The NFW content in the dough gradually increased as the addition of PSPF increased and the particle size decreased. When the amount of PSPF exceeded 10%, the content of NFW in PSPD started to be significantly higher than that of WFD (*p* < 0.05). When the amount of PSPF exceeded 10%, the water content of NFW in PSPD started to be significantly higher than that of WFD (*p* < 0.05). Kerch et al. [37] suggested that NFW was the tightly bound water in dough, which had a low freezing point and did not freeze, even at −40 °C. Therefore, our results suggested that the addition of PSPF lowered the freezing point temperature of the dough, inhibited the formation of ice crystals during freezing and the growth of ice crystals during freezing and storage, and improved the freezing stability of the dough. It is reasonable to predict that smaller PSPF particle sizes would be more effective in protecting the structure and texture of the dough from damage and preventing dough cracking, thus improving the storage stability of the dough and potentially extending its shelf life [38].

### 3.7. LF-NMR

Water is an essential ingredient in dough, and water mobility and its distribution play vital roles in the rheological properties of dough. In this study, the water content and distribution of the dough were confirmed by the T_2_ (transverse relaxation time) obtained from LF-NMR [39]. There were three peaks of T_21_ (0.4–2 ms), T_22_ (4–20 ms), and T_23_ (40–200 ms), which represented bound water (BW), immobilized water (IW), and free water (FW) in the dough, respectively. The peak area percentages A_21_, A_22_, and A_23_, corresponding to T_21_, T_22_, and T_23_, respectively, expressed the relative water content of each part [40].

The peak areas of the different doughs are shown in Table 6. With the addition of PSPF increasing from 0 to 20%, the A_21_ of the doughs increased, whereas the A_22_ decreased. This proves that the IW in the dough was converted into BW with the addition of PSPF, and the degree of conversion was positively correlated with the amount of PSPF. Conversely, the content of cellulose in PSPF was higher than that in WF. Cellulose contains a large amount of hydroxyl groups. The hydroxyl groups can interact with water through hydrogen bonds, enhancing the system’s water-binding ability [28]. However, it is believed that PSP contains a high amount of gelatinized starch, which improves the combination of the components with water molecules. With the substitution of PSPF, the binding capacity of dough and water was enhanced. This result is consistent with those of Liu, Yang, Zhao, and Zhang [41]. Furthermore, when the added amount of PSPF was the same, as the particle size of the PSPF decreased, the A_21_ of the dough initially increased and then tended to level off (or slightly decrease). Therefore, a decrease in PSPF particle size promoted the conversion of IW to BW in the dough. Because the particles of PSPF became smaller, the specific surface area of PSPF increased, and the number of hydrogen bonds that can combine with water on its surface also increased [42]. Lastly, small amounts of FW were detected in all samples. These may be water that is free on the surface of the dough and not adsorbed or bound to the dough.

### 3.8. Correlation Analysis

The correlation analysis of the rheological properties and water distribution results is shown in Table 7. WA showed a highly significant negative correlation (*p* < 0.01) with ST, Bw, and FzW content and a significant positive correlation (*p* < 0.05) with A_21_. These results suggest that PSPF enhanced the binding of the dough to water molecules by increasing the number of hydrophilic groups in the dough, thereby reducing the mobility of the water. This further stabilized the dough and prevented its icing. DT showed a highly significant negative correlation (*p* < 0.01) with A_22_, FzW, and ΔH. These results suggest that the high moisture fluidity of the dough facilitated the rapid formation of the gluten network. A_21_ showed highly significant negative correlations (*p* < 0.01) with ΔH and FzW. A_22_ showed highly significant positive correlations (*p* < 0.01) with ΔH and FzW. ΔH showed highly significant positive correlations (*p* < 0.01) with FzW. The regression curves for FW content measured by DSC and A_22_ measured by LF-NMR showed the same trend, confirming that LF-NMR and DSC could be used to determine the consistency of the moisture state in PSPD.

## 4. Conclusions

In this study, different particle sizes and addition levels of PSPF were added to doughs, which may have affected the moisture, starch, protein, and cellulose contents of the doughs. We provided evidence that such an influence resulted in changes in the rheological properties and moisture distribution of the PSPD. PSPF decreased the viscosity of the dough, but the pasting characteristics increased as the size of the PSPF decreased. The water absorption of the dough increased by PSPF addition, which resulted in the changes in dough development time, stabilization time, and bandwidth. Dynamic rheology showed that PSPF conferred greater dough elasticity. The original microstructure of the dough was disrupted, and a porous and dense dough structure was formed by the high addition levels and low particle sizes of PSPF. DSC and LF-NMR results showed that PSPF converted immobilized water and FzW to bound water and NFW in the dough, providing the dough with better processing and storage characteristics, and the decrease in PSPF particle size facilitated these processes. These results will provide important guidance for the development of different PSP flour-based products. The effects of PSPF on the active ingredients in the dough will be further investigated in the future.

## Figures and Tables

**Figure 1 foods-12-00398-f001:**
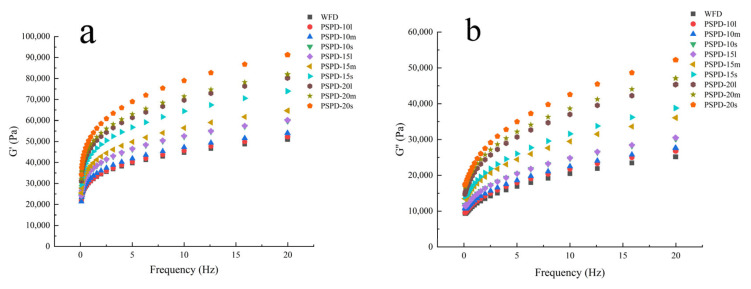
Mechanical spectra G’ (**a**) and G’’ (**b**) of WFD and PSPD samples with different levels (10%, 15%, and 20%) and different particle sizes (l, m, s) of PSPF.

**Figure 2 foods-12-00398-f002:**
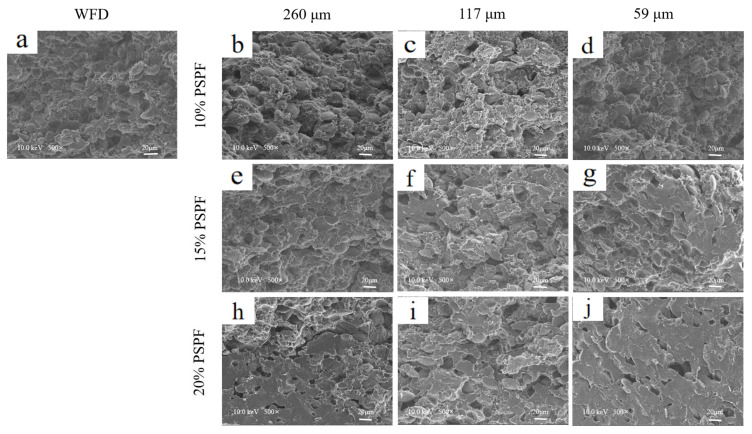
SEM images of the cross-sections of WFD (**a**) and PSPD (**b**–**j**) with different addition levels (10%, 15%, and 20%) and particle size (260 μm, 117 μm, and 59 μm) of PSPF.

**Table 1 foods-12-00398-t001:** Particle size distribution and nutrient distributions of WF and PSPF sieved through different sieves.

Sample	Size (D_4,3_ μm)	Moisture (%)	Starch (g/100 g)	Protein (g/100 g)	Cellulose (g/100 g)
WF	80 ± 0 c	14.25 ± 0.14 a	66.22 ± 0.77 a	17.42 ± 0.60 a	1.50 ± 0.07 d
PSPF 100 mesh	260 ± 1 a	6.76 ± 0.11 d	44.17 ± 0.09 b	3.42 ± 0.11 b	2.77 ± 0.01 a
140 mesh	117 ± 2 b	7.42 ± 0.07 c	41.20 ± 0.54 c	3.14 ± 0.13 c	2.46 ± 0.38 b
180 mesh	59 ± 1 d	8.02 ± 0.42 b	38.29 ± 1.20 d	2.65 ± 0.11 d	2.07 ± 0.17 c

Values are the mean ± standard deviation (SD). WF, wheat flour; PSPF, purple sweet potato flour. Different letters in the same column indicate significant differences (*p* < 0.05).

**Table 2 foods-12-00398-t002:** Pasting properties of formulated flours influenced by different particle size distributions and addition levels.

Sample	PV (cP)	HV (cP)	BD (cP)	FV (cP)	SB (cP)
WF	2045.33 ± 79.13 a	1334.33 ± 61.06 a	711.00 ± 18.08 a	2449.67 ± 86.09 a	1115.33 ± 86.31 a
FF-10l	1336.00 ± 53.45 bz	849.00 ± 42.88 by	487.00 ± 11.13 az	1670.67 ± 63.31 bz	821.67 ± 21.12 bz
FF-10m	1617.00 ± 157.64 by	1044.67 ± 97.12 bx	572.33 ± 60.71 ay	1973.33 ± 159.17 by	928.67 ± 62.96 by
FF-10s	2013.67 ± 76.35 bx	1058.67 ± 63.10 bx	955.00 ± 118.83 ax	2011.00 ± 116.05 bx	952.33 ± 57.27 bx
FF-15l	1220.67 ± 85.94 cz	793.33 ± 72.59 cy	427.33 ± 21.22 bz	1510.00 ± 96.16 cz	716.67 ± 30.92 cz
FF-15m	1473.67 ± 93.14 cy	802.67 ± 85.70 cx	671.00 ± 41.38 by	1553.00 ± 147.25 cy	750.33 ± 62.04 cy
FF-15s	1815.33 ± 44.41 cx	951.00 ± 27.49 cx	864.33 ± 49.09 bx	1834.00 ± 32.90 cx	883.00 ± 11.35 cx
FF-20l	1162.67 ± 33.85 dz	752.33 ± 31.34 cy	410.33 ± 19.65 cz	1474.00 ± 43.71 cz	721.67 ± 22.50 cz
FF-20m	1380.67 ± 76.78 dy	906.00 ± 67.08 cx	483.67 ± 18.77 cy	1717.33 ± 115.39 cy	811.33 ± 49.16 cy
FF-20s	1480.00 ± 75.62 dx	940.67 ± 62.69 cx	539.33 ± 15.56 cx	1804.67 ± 109.46 cx	864.00 ± 46.77 cx
Two-way ANOVA *p* value					
Factor I: Addition level	<0.001	<0.001	<0.001	<0.001	<0.001
Factor II: Particle size	<0.001	<0.001	<0.001	<0.001	<0.001
Factor I × Factor II	<0.001	0.030	<0.001	0.020	ns

PV, peak viscosity; HV, holding viscosity; BD, breakdown; FV, final viscosity; SB, setback; FF-10–20 refer to formula flours with 10%, 15%, and 20% PSPF substitution; l, m, s refer to large size (260 μm), medium size (117 μm), and small size (59 μm) of PSPF particle; a–d are the values obtained from the samples with the same particle size distribution, and different addition levels were significantly (*p* < 0.05) different; x–z, for the same parameter, are the values obtained from samples with the same addition level, and different particle size distributions were significantly (*p* < 0.05) different; ns, non-significant (*p* ≥ 0.05).

**Table 3 foods-12-00398-t003:** Mixing properties of formulated flours influenced by different particle size distributions and addition levels of PSPF.

Sample	WA (%)	DT (min)	ST (min)	Bw (mNm)
WF	57.47 ± 0.23 c	2.03 ± 0.06 c	1.30 ± 0.00 a	25.33 ± 0.58 a
FF-10l	57.13 ± 0.12 bz	2.47 ± 0.12 by	1.07 ± 0.06 bx	28.33 ± 0.58 ax
FF-10m	58.30 ± 0.00 by	2.70 ± 0.17 bx	0.93 ± 0.12 by	21.00 ± 1.00 ay
FF-10s	59.03 ± 0.23 bx	2.57 ± 0.06 by	0.67 ± 0.12 bz	22.33 ± 1.53 ay
FF-15l	57.00 ± 0.00 bz	2.57 ± 0.06 by	1.07 ± 0.06 bx	30.67 ± 2.89 ax
FF-15m	58.83 ± 0.46 by	2.83 ± 0.12 bx	1.00 ± 0.20 by	21.67 ± 1.53 ay
FF-15s	59.23 ± 0.11 bx	2.60 ± 0.00 by	0.60 ± 0.00 bz	21.67 ± 1.15 ay
FF-20l	57.80 ± 0.00 az	2.77 ± 0.06 ay	1.07 ± 0.21 bx	25.67 ± 0.58 bx
FF-20m	58.83 ± 0.46 ay	2.97 ± 0.06 ax	0.87 ± 0.15 by	20.33 ± 0.58 by
FF-20s	59.63 ± 0.06 ax	2.83 ± 0.06 ay	0.67 ± 0.06 bz	21.33 ± 1.15 by
Two-way ANOVA *p* value				
Factor I: PSPP addition	<0.001	<0.001	ns	<0.001
Factor II: Particle size	<0.001	<0.001	<0.001	0.011
Factor I × Factor II	<0.001	ns	ns	0.042

WA, water absorption; DT, dough development time; ST, dough stability time; Bw, bandwidth; when the samples at the same particle size distributions with different addition level, a–d for the same parameter with different superscripts means that differences were significant (*p* < 0.05); when the samples at the same addition level with different particle size distributions, x–z for the same parameter with different superscripts means that differences were significant (*p* < 0.05); ns, non-significant (*p* ≥ 0.05); the same below.

**Table 4 foods-12-00398-t004:** Effects of particle sizes and addition levels on power law models in dough frequency scan tests.

Sample	G′ = K″·ω^n′^			G″ = K″·ω^n″^		
	K′ (Pa s^n′^)	n′	R^2^	K″ (Pa s^n″^)	n″	R^2^
WFD	31,356.33 ± 118.72 g	0.156 ± 0.002 e	0.996	12,309.32 ± 213.02 h	0.223 ± 0.008 c	0.972
PSPD-10l	31,529.03 ± 126.82 g	0.162 ± 0.002 de	0.997	12,900.99 ± 208.74 g	0.229 ± 0.008 b	0.977
PSPD-10m	32,606.38 ± 114.55 f	0.163 ± 0.001 d	0.998	13,755.12 ± 256.69 f	0.216 ± 0.009 d	0.965
PSPD-10s	35,939.03 ± 126.06 e	0.165 ± 0.002 d	0.998	15,035.27 ± 259.05 e	0.217 ± 0.008 d	0.970
PSPD-15l	36,213.81 ± 134.36 e	0.164 ± 0.002 d	0.997	14,955.14 ± 263.36 e	0.222 ± 0.009 c	0.971
PSPD-15m	38,621.43 ± 138.76 d	0.167 ± 0.002 c	0.997	17,820.99 ± 286.04 d	0.221 ± 0.008 c	0.975
PSPD-15s	43,934.30 ± 169.17 c	0.168 ± 0.001 c	0.996	18,777.33 ± 311.37 d	0.228 ± 0.008 b	0.976
PSPD-20l	47,204.90 ± 179.05 b	0.171 ± 0.003 bc	0.997	21,572.31 ± 258.38 c	0.237 ± 0.006 a	0.989
PSPD-20m	48,868.44 ± 196.16 b	0.174 ± 0.002 b	0.998	23,191.29 ± 320.38 b	0.224 ± 0.007 c	0.982
PSPD-20s	52,526.19 ± 213.71 a	0.179 ± 0.004 a	0.997	24,434.93 ± 344.08 a	0.242 ± 0.007 a	0.985

Different letters in the same column indicate significant differences (*p* < 0.05).

**Table 5 foods-12-00398-t005:** Melting enthalpy, freezable water, and non-freezable water content of PSPD.

PSPF (%)	260 μm			117 μm			59 μm		
	ΔH (J·g^−1^)	FzW (%)	NFW (%)	ΔH (J·g^−1^)	FzW (%)	NFW (%)	ΔH (J·g^−1^)	FzW (%)	NFW (%)
0	52.85 ± 1.22 a	43.20 ± 0.87 a	56.82 ± 0.74 d	52.85 ± 1.22 a	43.20 ± 0.87 a	56.82 ± 0.47 c	52.85 ± 1.22 a	43.20 ± 0.87 a	56.82 ± 0.47 d
10	47.46 ± 0.98 b	41.43 ± 0.50 b	58.57 ± 1.24 c	47.09 ± 0.24 b	39.27 ± 1.20 b	60.73 ± 2.13 b	46.24 ± 0.22 b	38.15 ± 1.08 b	61.85 ± 0.43 c
15	46.23 ± 1.76 bc	40.04 ± 0.89 c	59.96 ± 2.20 b	44.33 ± 0.38 c	39.09 ± 0.60 b	60.90 ± 1.59 b	41.07 ± 1.67 c	37.02 ± 0.54 bc	63.01 ± 1.12 b
20	43.71 ± 0.63 c	38.19 ± 1.16 d	61.80 ± 0.16 a	41.06 ± 0.99 d	36.60 ± 0.69 c	63.35 ± 1.72 a	39.32 ± 1.07 d	35.44 ± 0.33 c	64.59 ± 1.10 a

ΔH, melting enthalpy; FzW, freezable water content; NFW, non-freezable water content. Different letters in the same column indicate significant differences (*p* < 0.05).

**Table 6 foods-12-00398-t006:** Fractions of signal amplitude of protons of formulated flour doughs influenced by different particle size distributions and addition levels detected by low-field ^1^H NMR.

PSPF (%)	Peak Area (%)								
	260 μm			117 μm			59 μm		
	A_21_	A_22_	A_23_	A_21_	A_22_	A_23_	A_21_	A_22_	A_23_
0	9.26 ± 0.05 d	90.37 ± 1.68 a	0.46 ± 0.03 c	9.26 ± 0.05 d	90.37 ± 1.68 a	0.46 ± 0.03 d	9.26 ± 0.05 d	90.37 ± 1.68 a	0.46 ± 0.03 b
10	12.07 ± 0.34 c	87.60 ± 1.05 b	0.39 ± 0.07 c	16.11 ± 0.24 c	83.04 ± 1.50 b	0.97 ± 0.01 b	16.85 ± 0.27 c	82.66 ± 2.43 b	0.39 ± 0.09 c
15	20.01 ± 0.57 b	77.12 ± 0.32 c	2.84 ± 0.09 a	23.54 ± 0.71 b	74.50 ± 0.77 c	1.93 ± 0.07 a	23.86 ± 0.21 b	75.59 ± 1.45 c	0.62 ± 0.03 a
20	24.82 ± 0.24 a	73.86 ± 0.98 d	1.33 ± 0.14 b	32.27 ± 0.30 a	66.95 ± 0.41 d	0.60 ± 0.06 c	31.90 ± 0.21 a	67.70 ± 1.09 d	0.36 ± 0.02 c

A_21_, bound water content; A_22_, immobilized water content; A_23_, free water content. The signal amplitude of protons in one gram. Different letters in the same column indicate significant differences (*p* < 0.05).

**Table 7 foods-12-00398-t007:** Correlation analysis.

	PV	HV	BD	FV	SB	WA	DT	ST	Bw	A_21_	A_22_	A_23_	ΔH	FzW	NFW
PV	1														
HV	0.863 **	1													
BD	0.884 **	0.527	1												
FV	0.876 **	0.998 **	0.552	1											
SB	0.889 **	0.987 **	0.583	0.995 **	1										
WA	0.316	0.049	0.490	0.069	0.098	1									
DT	−0.566	−0.690 *	−0.308	−0.690 *	−0.685 *	0.539	1								
ST	−0.21	0.128	−0.475	0.096	0.049	−0.838 **	−0.507	1							
Bw	−0.376	−0.231	−0.426	−0.244	−0.262	−0.881 **	−0.478	0.598	1						
A_21_	−0.431	−0.505	−0.252	−0.514	−0.522	0.640 *	0.852 **	−0.54	−0.485	1					
A_22_	0.489	0.565	0.294	0.579	0.593	−0.581	−0.855 **	0.490	0.411	−0.993 **	1				
A_23_	−0.524	−0.536	−0.390	−0.580	−0.636 *	−0.402	0.168	0.337	0.479	0.059	−0.175	1			
ΔH	0.382	0.570	0.111	0.557	0.534	−0.701 *	−0.838 **	0.741 *	0.473	−0.928 **	0.910 **	0.046	1		
FzW	0.198	0.395	−0.038	0.383	0.363	−0.812 **	−0.820 **	0.833 **	0.613	−0.897 **	0.865 **	0.144	0.949 **	1	
NFW	−0.193	−0.391	0.042	−0.379	−0.359	0.814 **	0.816 **	−0.835 **	−0.612	0.895 **	−0.863 **	−0.146	−0.949 **	−1 **	1

* indicates significance at *p* < 0.05; ** indicates significance at *p* < 0.01.

## Data Availability

Data is contained within the article.

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
