# Peer review of "Influences of Particle Size and Addition Level on the Rheological Properties and Water Mobility of Purple Sweet Potato Dough"

_foods, 2023, doi:10.3390/foods12020398_

Round 1
Reviewer 1 Report
This manuscript characterized the effects of purple sweet potato flour on the pasting properties, farinograph characteristics, dynamic rheological properties, microstructure, and water state of the dough. They found that the adverse effects of purple sweet potato flour on the dough could be reduced by reducing the particle size, thus suggesting an effective solution for enhancing the quality of purple sweet potato dough. Overall, this is a well-conducted work, it applied proper methods and the findings are interesting to the audience of Foods. I recommend revisions to address some issues as follows:
(1) line 41-46: Previous studies [11] [12] were cited here to show that changing the particle size is beneficial for improving dough quality, but the theoretical discussion was not shown. A proper discussion is needed here.
(2) Line 50: ”The process of dough formation is mainly dependent on the contents of water molecules and macromolecules, such as proteins and starch, in the dough”. This part is not clear- please clarify.
(3) Line 64: The source of the purple potatoes needs to be supplemented with a more detailed description, including but not limited to the origin, storage time, and selection criteria.
(4) Line 67: “... and dehydrated in a cabinet dryer (101-3EBS, Yongguang Medical Instruments
Co., Ltd., Beijing, China) at 60° C until the moisture was maintained at 6–10%”. How could the authors control the water content around 6-10%? Please describe it in detail.
(5) Line 128-133: Is this a new method? If not, please add a reference.
(6) Line 262-278: If the gluten content decreased with PSPF substitution, why the 20% PSPF substitution dough showed higher elasticity?
(7) English needs to be polished.
Author Response
Response to Reviewers’ comments
Dear reviewer:
We thank you very much for your great comments and appreciate you for your positive remarks on our MS “Overall, this is a well-conducted work, it applied proper methods and the findings are interesting to the audience of Foods. “
- line 41-46: Previous studies [11] [12] were cited here to show that changing the particle size is beneficial for improving dough quality, but the theoretical discussion was not shown. A proper discussion is needed here.
Response: We thank the reviewer’s valuable suggestion. We have added discussions as:" Changes in the particle size of the flour affect its specific surface area and porosity, which in turn affects the packing density, the number of binding sites, and the exposure of active ingredients of the flour, leading to an impact on both the formation and nutritional value of the dough."(line 43-46)
- Line 50: ”The process of dough formation is mainly dependent on the contents of water molecules and macromolecules, such as proteins and starch, in the dough”. This part is not clear- please clarify.
Response: We are grateful to the reviewer for the careful review. We have revised this sentence to “The process of dough formation is mainly dependent on the combinations of water molecules and macromolecules, such as proteins and starch, in the dough” . (line 54-56)
- Line 64: The source of the purple potatoes needs to be supplemented with a more detailed description, including but not limited to the origin, storage time, and selection criteria.
Response: We are very grateful to the reviewer for this suggestion. We have added detailed informations about the source of purple sweet potatoes as:” The PSP variety Mianzishu 9 was supplied by the Hunan Academy of Agricultural Sciences and planted in the experimental field of the Hunan Crop Research Institute. After harvesting in October 2022, fresh PSPes free of rot, sprouting, diseases and pests were selected and sent to our laboratory immediately.” ( line 69-72)
- Line 67: “... and dehydrated in a cabinet dryer (101-3EBS, Yongguang Medical Instruments
Co., Ltd., Beijing, China) at 60° C until the moisture was maintained at 6–10%”. How could the authors control the water content around 6-10%? Please describe it in detail.
Response: We thank the reviewer’s valuable suggestion. We apologise for the lack of descriptions of moisture monitoring methods for tubers here. “A tuber was removed and its moisture content was determined every half hour using a halogen rapid moisture analyser (JH-HS, Yixind Instrument Co., Ltd., Jiangsu, China), until the moisture was maintained at 6–10%” was added here now. (line 75 -77)
- Line 128-133: Is this a new method? If not, please add a reference.
Response: This is a cited method. We are very sorry for the omission of the reference. The reference has now been added as ref. [20]. (line 138-139)
- Line 262-278: If the gluten content decreased with PSPF substitution, why the 20% PSPF substitution dough showed higher elasticity?
Response: We thank the reviewer’s valuable suggestion. The gluten content indeed decreased with the increase of PSPF addition level, but the cellulose and amylose contents were increased. This is because that cellulose could interact with proteins by hydrogen bond between hydroxyl gourps of cellulose and the carbonyl groups of protein residues (DOI:.org/10.1016/j.foodchem.2017.05.052), which enhanced the network of dough and increased the elasticity of dough. Additionally, the amylose aggregates were also filled into the hole of the gluten network, which also resulted in the higher elasticity. Similar results were obtained with other reports (DOI: 10.3390/foods8100519; DOI: 10.3390/foods9030298).
- English needs to be polished
Response: We are very grateful for the reviewer’s suggestion. The manuscript has been thoroughly polished now. Please review the revised version we have uploaded.

Reviewer 2 Report
The manuscript titled "Influences of particle size and addition level on the rheological properties and water mobility of purple sweet potato dough" deals within the scope of the Foods Journal, by investigating an interesting topic of research. The quality of the presented research work is very good and represents valuable research results on the application of purple sweet potato as a wheat flour substitute.
Therefore, I would suggest minor corrections.
Below are some comments to help you revise the manuscript.
Corrections to be made:
Lines 22, 61, 401: The authors should not highlight pasta as the only possible product developed, as the results of this study may be helpful in providing valuable information on the development of some other flour-based products with purple sweet potato.
Line 107: To the best of my knowledge, Micro-doughLAB is only a similar device to the Farinograph, but not a real Farinograph. Therefore, it is sufficient to call it "Micro-doughLAB". Please, correct this in the rest of the text accordingly. In addition, please check the manufacturer of the device.
Line 110: The authors should also check if the water absorption was estimated to a maximum torque of 100 mNm or to a consistency of 500 BU.
Line 173 (Table 1): Please check the results in Table 1. When "total carbohydrates" is mentioned, it usually means all carbohydrates, including the starch content.
Author Response
Response to Reviewers’ comments
Dear reviewer:
We thank you very much for your great comments and appreciate you for your positive remarks on our MS “The quality of the presented research work is very good and represents valuable research results on the application of purple sweet potato as a wheat flour substitute. “
- Lines 22, 61, 401: The authors should not highlight pasta as the only possible product developed, as the results of this study may be helpful in providing valuable information on the development of some other flour-based products with purple sweet potato.
Response: We thank the reviewer’s valuable suggestion. As suggested by the reviewer, “pasta products” in the manuscript have been replaced with “flour-based products”. (line 22, 66, 426 )
- Line 107: To the best of my knowledge, Micro-doughLAB is only a similar device to the Farinograph, but not a real Farinograph. Therefore, it is sufficient to call it "Micro-doughLAB". Please, correct this in the rest of the text accordingly. In addition, please check the manufacturer of the device.
Response: We are very grateful to the reviewers for this professional suggestion. We have revised the “Farinograph” in the manuscript to “Micro-doughLAB”, and have revised the “Farinograph characteristics” in the manuscript to “Mixing properties”. (line 14, 63, 115, 116, 117, 128, 228, 230, 236 )
- Line 110: The authors should also check if the water absorption was estimated to a maximum torque of 100 mNm or to a consistency of 500 BU.
Response: We are very grateful for the reviewer’s suggestion. This sentence was not expressed very clearly.A more detailed description of the method was revised as "The water absorption was determined to reach a peak of 500 BU using a 3.50 ± 0.01 g sample at a 14% moisture basis after moisture correction. For 15 min, samples were mixed at a speed of 63 rpm and at a temperature of 30 ℃. Dough development time, dough stabilisation time, and bandwidth were calculated using the mixing curve. " (line 118-122)
- Line 173 (Table 1): Please check the results in Table 1. When "total carbohydrates" is mentioned, it usually means all carbohydrates, including the starch content
Response: We are very grateful to the reviewers for this professional corrections. In the revised manuscript, the total carbohydrates content was removed from Table 1. As mentioned by reviewer that starch is included in all carbohydrates, and the contents of the starch and cellulose were both detected, which were already supported for our conclusion. (line 188 )
